# Comparative Analysis of Selected Geostatistical Methods for Bottom Surface Modeling

**DOI:** 10.3390/s23083941

**Published:** 2023-04-13

**Authors:** Patryk Biernacik, Witold Kazimierski, Marta Włodarczyk-Sielicka

**Affiliations:** 1Faculty of Navigation, Maritime University of Szczecin, Waly Chrobrego 1-2, 70-500 Szczecin, Poland; p.biernacik@pm.szczecin.pl; 2Marine Technology Ltd., 81-521 Gdynia, Poland; m.wlodarczyk@marinetechnology.pl

**Keywords:** seabed, DTM, DBM, hydrography, bathymetric data, spatial interpolation, geostatistical methods, bottom modelling, kriging

## Abstract

Digital bottom models are commonly used in many fields of human activity, such as navigation, harbor and offshore technologies, or environmental studies. In many cases, they are the basis for further analysis. They are prepared based on bathymetric measurements, which in many cases have the form of large datasets. Therefore, various interpolation methods are used for calculating these models. In this paper, we present the analysis in which we compared selected methods for bottom surface modeling with a particular focus on geostatistical methods. The aim was to compare five variants of Kriging and three deterministic methods. The research was performed with real data acquired with the use of an autonomous surface vehicle. The collected bathymetric data were reduced (from about 5 million points to about 500 points) and analyzed. A ranking approach was proposed to perform a complex and comprehensive analysis integrating typically used error statistics—mean absolute error, standard deviation and root mean square error. This approach allowed the inclusion of various views on methods of assessment while integrating various metrics and factors. The results show that geostatistical methods perform very well. The best results were achieved with the modifications of classical Kriging methods, which are disjunctive Kriging and empirical Bayesian Kriging. For these two methods, good statistics were calculated compared to other methods (for example, the mean absolute error for disjunctive Kriging was 0.23 m, while for universal Kriging and simple Kriging, it was 0.26 m and 0.25 m, respectively). However, it is worth mentioning that interpolation based on radial basis function in some cases is comparable to Kriging in its performance. The proposed ranking approach was proven to be useful and can be utilized in the future for choosing and comparing DBMs, mostly in mapping and analyzing seabed changes, for example in dredging operations. The research will be used during the implementation of the new multidimensional and multitemporal coastal zone monitoring system using autonomous, unmanned floating platforms. The prototype of this system is at the design stage and is expected to be implemented.

## 1. Introduction

Getting to know the bottom of a water region is essential not only for the safety of navigation but also for environmental and industrial issues. Bathymetric surveys are made for construction or dredging in ports, designing the routes of cables and pipelines, discovering minerals or observing and controlling environmental changes (erosion, siltation). An example of using bathymetric data for environmental analysis can be seen in [1]. The authors investigated the spatiotemporal variations of water level fluctuations and tidal amplification. In addition, the authors developed a conceptual model of interactions between the river, tide and morphological evolution for shallow systems (it is worth emphasizing that the data collected for these studies also refer to the shallows). For those analyses, Digital Bottom Models (DBM) can be prepared based on survey data, showing depth distribution [2]. They can be useful for creating both—simple and complex analyses, such as generating prediction maps for various phenomena at the bottom and in the water column [3].

DBMs are built using interpolators to estimate the values of unmeasured points. Simultaneously, interpolation is one of the methods of spatial data analysis and is considered the most important in geoinformatics [4]. In general, spatial data analysis involves all kinds of transformations and calculations aimed at the appropriate preparation of spatial information for decision-making and scientific purposes [4]. The results are not only dependent on the data’s geolocation (changing location changes the results of analysis) but also on the data’s accuracy, reliability or used technologies [5].

One of the types of spatial analysis is geostatistics, which has been utilized in this research. The key to geostatistics is to analyze the data with statistical methods, however, taking into consideration not only the features of the data but also the location in space [6]. In such cases, statistical dependences among geographically distributed features are elaborated. Geostatistics allows to model and analyze continuous features, for which the feature has a determined (but unknown) value in every point of the analyzed area [7]. Bathymetry is an example of such data. The most popular geostatistical interpolation method is called Kriging. It is regarded as an optimal method of spatial prediction that weighs the surrounding measured points to calculate a prediction for an unknown location [8,9]. It is widely used for digital spatial data modeling, as well as bathymetrical data modeling in many variants, e.g., in [10,11].

In fact, in the literature, we can find various interpolation techniques for digital bottom modeling, but also for other environmental factors. For example, in [12], the use of various interpolation techniques is tested in one of the use cases. In general, various algorithms can be used for DEM modeling with various types of spatial data. A fine example of this can be found in [12,13,14], which shows a variety of methods. Inverse distance weighting (IDW) is one of the most popular methods adopted by geoscientists and geographers and has been implemented in many GIS packages. An example of using it for bathymetric data from a multi-beam echosounder can be found in [15]. Radial basis function interpolation (RBF) is an accurate deterministic spatial interpolation method that provides accurate prediction surfaces, which is beneficial for approximating surfaces [16]. Global Polynomial Interpolation (GPI) fits a smooth surface that is defined by a mathematical function to the input sample points [17]. Each of them has advantages and disadvantages. Many of them are also used for bottom modeling. Apart from the already mentioned IDW, analyzed also in [18], the natural neighbor algorithm is found to be used for this purpose, as in [19] or the spline method [20]. However, the most popular method used for bathymetric data modeling is Kriging, sometimes in a modified version. In many cases, Kriging methods provide the best performance in terms of prediction errors. In [19] a comparison of Kriging with the natural neighbor method is presented, showing that in various environments, the results are different. The method to improve the interpolation accuracy of the Digital Bottom Model by interpolating the submarine topography within the zone is provided, along with a suitable method for a zone. As a result, it was proven that for complex entire areas, Kriging provides more accurate models. In [18], a comparison of Kriging with IDW is provided using the recorded data of groundwater depth, again showing the Kriging advantage in that case. Another comparative study is given in [20], where the methods of inverse distance weighted interpolation, spline function and ordinary Kriging interpolation are examined for marine environmental monitoring. It was shown that if the sampling data spots are rare, ordinary Kriging using local spatial fitting is the preferred interpolation method, as it is more precise than others and can get better interpolation results. The Kriging method was also chosen for seabed sediment maps in [21] and for bathymetric mapping of shallow water. It can be seen that Kriging is often chosen. It is flexible and includes many parameters, which can be both—positive, as it allows precise modeling, and negative, as using Kriging forces a detailed and relatively complicated data analysis, searching trends and modeling [22,23,24]. This causes more workload than in the case of other methods, but on the other hand, the faster analysis does not have to be more accurate.

There were also some trials to improve the ordinary Kriging method, by providing modifications to the original algorithm. For example, in [25], two variants of Kriging were investigated—ordinary Kriging and universal Kriging for 3D Bathymetric models. In [21], the same authors analyzed deeply the parameters to demonstrate that the level of accuracy that can be achieved depends crucially on the choice of one of these parameters, that is, the mathematical model of the semivariogram. In [26] authors propose a new modification, called partitioned ordinary Kriging, improving its applicability for online updates for autonomous vehicles. On the other hand, another modification called growing radius Kriging is proposed [24]. The purpose of the optimization was to significantly decrease computation time while maintaining the highest possible accuracy of the created model. It can thus be noticed that the Kriging method is willingly analyzed in literature for digital bottom modeling and for comparison with other methods. Usually, two or three methods are compared, and one or two types of Kriging are used. One can, however, notice gaps in the existing literature, which gives room for further research. A complex and comprehensive approach to geostatistical DBM validation can be an added value, as the publications show mainly the selected aspects of the problem, comparing Kriging in general to other methods or going deeper into the Kriging itself. Therefore, the research questions we would like to answer are—what is the influence of various Kriging factors on the DBM, also comparing it to other methods and how to perform a complex and effective verification of the methods?

In any kind of analysis that is relying on a comparison of the accuracy and efficiency of spatial digital models, the key aspect is to choose proper methods and metrics for assessment. They are usually based on geostatistical analysis for some selected points or entire datasets in the model. Typically used metrics are root mean square error (RMSE) and MAE (mean absolute error) [18,19,20,26] or root mean square standardized error (RMSSE) [15,21]. Sometimes other statistics are used, such as minimum and maximum values [10,25] or standard deviation [19,21,25]. In some cases, authors provide their own proposals for the comparison metrics. For example, in [24] the percentage of blank nodes as well as the computation time appeared. In [11], the correlation coefficient is used together with the index of agreement. In [21], prediction standard error is used. In many cases [10,11,25], also cross-validation is used. In [27], the author provides a complex approach to accuracy, comparing first the technologies and then analyzing geostatistical errors. Finally, the Monte Carlo method is proposed to estimate horizontal errors. It can be concluded that many approaches are met for accurate assessment; however, there is still room for rational, complex assessment methods by combining a few factors.

In this study, we have two goals. First, we would like to present a comprehensive comparison of geostatistical and deterministic methods for bathymetric data by performing interpolation surfaces. We analyzed five variants of the Kriging method and three deterministic methods—IDW, RBF and GPI. Especially the last two are not very often met in publications, as well as a variety of Kriging. Secondly, we would like to perform complex verification with the use of various statistics and factors. For this reason, we propose a verification ranking, which allows a comparison of various aspects of the models. The idea of a comprehensive analysis stems from the publications cited earlier—until now, no analysis has been realized that took into account so many factors simultaneously.

## 2. Materials and Methods

### 2.1. Research Methodology

Geostatistical modeling consists of several steps. Figure 1 shows the diagram of the methodology used in this work.

The organization plan and collection of the data are described in part 2. Exploratory Spatial Data Analysis (ESDA) is performed to determine basic statistical parameters (the mean, standard deviation, coefficient of variation and maximum and minimum values), identify outliers and calculate the data distribution [28]. Structural analysis, or spatial continuity analysis, is made to determine the correlation between the data. Based on the semivariogram function, the dependency of distance and direction between these data is checked. It is necessary to model a variogram function properly. After the completion of verification, the next step is choosing one of the interpolation methods and making a prediction. Then the predictions are compared and rated to draw conclusions.

The steps from ESDA to assessing predictions were done in ArcGIS software. ArcGIS has a dedicated extension for geostatistical analysis. This research was not only a geospatial and visualization platform but was also used to create statistics for comparative analysis.

### 2.2. Study Area

Bathymetric data were collected during the survey mission on 21 March 2022 in Lake Klodno on the side of Zawory village, Pomorskie Voivodeship, Poland (red dot in Figure 2). The survey was the test of the unmanned vessel HydroDron-1 at the same time (Figure 3). HydroDron-1 is an innovative craft because of its autonomy—it is a product of Marine Technology, the company dealing with collecting and processing bathymetric data. The equipment of HydroDron-1 includes an integrated bathymetric and sonar system, data acquisition computer, inertial navigation system, profilometer and sound speed meter in the water, single-beam dual-frequency sonar, single-beam high-frequency sonar, LiDAR, radar and GNSS receiver. Data for analysis were collected using the interferometric sonar PING 3DSS-DX-450 (Figure 3). The depth measurement involves determining the time describing the way of the hydroacoustic wave from the transducer to the bottom along with the return to the transducer. To determine the value of depth, it is also necessary to know values such as the time of the wave pulse, the immersion of the transducer and the speed of sound in the water, which were measured in this integrated system [29].

Data were measured from seven measurement profiles in total, comprising two separate measurement missions; two types of profiles were created—one mission for profiles parallel to the shore and one mission for profiles perpendicular to the shore. All data were surveyed in the WGS 1984 UTM coordinate system zone 34N (EPSG code: 32634). Data from profiles perpendicular to the shore were used for analysis due to the smaller number of points—5 million, while data from parallel profiles contain more than 11 million points. The data from the parallel profiles were left for possible measurement control. Figure 4 presents raw bathymetric data and designed profiles in the HYPACK software. The spatial resolution for these observations is 0.5 × 0.5 m. It should be noted here that Figure 4 is only a preview illustration. The target interpolation surfaces analyzed in the research have different resolutions.

Figure 5 and Figure 6 show histograms for raw data with general statistics. The count of points has decreased because ArcMap has removed overlapping points by default. The histogram provides an opportunity to describe the variable. It is a graphical representation of the distribution of the data, i.e., the graph shows how often the observed values fall into predetermined intervals or classes. The horizontal axis of the histogram indicates the specific interval in which the values are combined, while the vertical axis represents the number of points (observations) in these intervals.

### 2.3. Interpolation Methods

In this work, we decided to also use selected deterministic methods, apart from the geostatistical methods. Deterministic methods are often compared with geostatistical methods to indicate whether geostatistical methods perform better for spatial data, e.g. [18,19,20,23]. Deterministic methods that were used for interpolation were Inverse Distance Weighted (IDW), Radial Basis Function (RBF) and Global Polynomial Interpolation (GPI). When it comes to geostatistical methods, a few kinds of Kriging were used: ordinary Kriging (OK), simple Kriging (SK), universal Kriging (UK), disjunctive Kriging (DK) and empirical Bayesian Kriging (EBK). Estimates of interpolation surfaces were made using ArcGIS software with the Geostatistical Analyst extension.

#### 2.3.1. Inverse Distance Weighted

Inverse Distance Weighted is a deterministic estimation method in which values at unmeasured points are determined by a linear combination of values at nearby measured points [18,30,31]. The main assumption is that the values of the weights are inversely proportional to the powers of the distance between the interpolation points and the measurement point [32]. The IDW function should be used when the set of points is dense enough to capture the range of local surface variation needed for analysis. If the sampling of input points is sparse or uneven, the results may not adequately represent the desired surface area [33]. The advantage of the inverse distance method is its simplicity and efficiency. In addition, IDW can handle extreme terrain changes, such as faults. Disadvantages include sensitivity to outliers—the average cannot be greater than the highest input value nor lower than the lowest. This makes the method unsuitable for creating ridges and valleys. IDW will also not work well for interpolating mountainous areas. The formula for the IDW method is presented below in Equation (1) [31,34]:(1)$${\hat{Z}}_{j}=\frac{\sum_{i}\frac{z_{i}}{{\left(h_{ij}+s\right)}^{p}}}{\sum_{i}\frac{1}{{\left(h_{ij}+s\right)}^{p}}},$$
where:

${\hat{Z}}_{j}$—interpolated value in point *j*,

$z_{i}$—measured values in points *I* (*i* = 1, …, *n*),

$h_{ij}$—distance between points *i* and *j*,

s—smoothing factor (*s* > 0),

p—weight exponent.

The value of the weight exponent is determined by minimizing the RMSE [33].

#### 2.3.2. Radial Basis Function

Radial Basis Function methods are exact interpolation techniques [35]. This means that the interpolation surface must pass through every point where the value is known. The following are five basis functions: thin plate spline (TPS), spline with tension (ST), completely regularized spline (CRS), multiquadric (MQ) (that were used in prediction) and inverse multiquadric (IMQ) [36]. The estimated values are based on a mathematical function that minimizes the total curvature. This causes the resulting surfaces to be smooth. Unlike IDW, RBF can determine values above the maximum and below the minimum of the measured values [17]. RBF gives good results for slightly varying surfaces, such as terrain with small differences in elevation. However, they will not be suitable for large differences in values or for data with large measurement uncertainty [2]. An important advantage of RBF is that it can be used for large data sets (like bathymetric data). The mathematical Equation (2) is given by [36]:(2)$$Z\left(x\right)\;\overset{n}{\underset{i-1}{\sum }}a_{i}f_{i}\left(x\right)+\overset{n}{\underset{i-1}{\sum }}b_{j}\varphi \left(d_{j}\right),$$
where:

*f*(*x*)—process of the function,

$\varphi \left(d\right)$—radial base function,

$d_{j}$—distance between the points of sampling and the predicted point *x*.

All equations for the previously mentioned five basis functions are given below in Equations (3)–(7) [37]:(3)$$TPS-\varphi \left(d\right)=c^{2}d^{2}\ln\left(cd\right)$$
(4)$$ST-\varphi \left(d\right)=ln\frac{cd}{2}+I_{0}\left(cd\right)+\gamma$$
(5)$$CRS-\varphi \left(d\right)=\ln\left(\frac{cd^{2}}{2}\right)+E_{1}{\left(cd\right)}^{2}+y$$
(6)$$MQ-\varphi \left(d\right)=(\sqrt{d^{2}+c^{2})}$$
(7)$$IMQ-\varphi \left(d\right)=({\sqrt{d^{2}+c^{2})}}^{-1},$$
where:

*d*—distance from sample to prediction location,

*c*—smoothing factor,

*I*_0_()—modified Bessel function,

*E*—Euler’s constant.

#### 2.3.3. Global Polynomial Interpolation

The Global Polynomial Interpolation method involves fitting a smooth surface, defined by a mathematical function (polynomial), to the data points of the input [22]. The method of global polynomial interpolation is classified as an inaccurate interpolator because the mathematical function rarely passes through all the actual measured points. Calculated surfaces are highly susceptible to outliers (extremely high and low values), especially at the edges [33]. This method can be used for trend surface analysis but will also work well in cases where the surface varies slowly from region to region.

#### 2.3.4. Kriging

The Kriging method is often referred to by the acronym B.L.U.E. (best linear unbiased estimator). Kriging is linear because its estimates are linear, weighted, and combinations of data [31]. The unbiasedness is due to the fact that the goal of Kriging is to obtain a mean error equal to 0, while “best” refers to the minimization of error variation. This minimization of error variation distinguishes Kriging from other methods. Equation (8) shows an estimation of unknown values at different points using different sets of weights [34]:(8)$$\hat{z}(x_{0})=\overset{n}{\underset{i=1}{\sum }}w_{i}z\left(x_{i}\right),$$
where:

$\hat{z}(x_{0})$—the unknown value at any point,

*w_i_*—sets of weights,

*z*(*x_i_*)—the value of the measured quantity at neighboring points.

There are a few Kriging methods (or types of Kriging methods), as was said at the beginning of this part. Simple Kriging assumes that the average is known and constant throughout the area. In ordinary Kriging, the average is treated as an unknown but fixed value. Universal Kriging assumes that the unknown local average changes gradually over the study area [38]. Disjunctive Kriging represents a form of nonlinear kriging, which in general offers an improvement over linear kriging methods yet does not require knowledge of the *n* = 1 joint probability distributions necessary for the conditional expectation [37]. Disjunctive Kriging mainly uses Hermite polynomials and Gaussian bivariate assumptions. It is not a very popular type of Kriging. Empirical Bayesian Kriging differs from classical Kriging methods. EBK considers the error introduced by the estimation of the semivariogram model. This is done by estimating, and then using, many semivariogram models rather than a single semivariogram. This process entails the following steps:A semivariogram model is estimated from the data.Using this semivariogram, a new value is simulated at each of the input data locations.A new semivariogram model is estimated from the simulated data. Weight for this semivariogram is then calculated using Bayes’ rule, which shows how likely the observed data can be generated from the semivariogram.

Steps 2 and 3 are repeated. With each repetition, the semivariogram estimated in Step 1 is used to simulate a new set of values at the input locations. The simulated data is used to estimate a new semivariogram model and its weight. Predictions and prediction standard errors are then produced at the unsampled locations using these weights [39].

### 2.4. Validation Methods

Before a final interpolation surface is created, it is necessary to verify how well the model predicts values at unknown locations. This is what cross-validation will be used for. Cross-validation uses all data to estimate models. It removes each data location one at a time and predicts the associated data value using the points left behind [37]. The quality of the comparison of results between several estimations can be assessed using estimation quality statistics. These include the following [8,38,40,41]:Mean error (ME);Root mean square error (RMSE);Average standard error (ASE);Mean standardized error (MSE);Root mean square standardized error (RMSSE).

All statistics are available for geostatistical methods. For deterministic methods, only ME and RMSE can be calculated. Optimally, the values of ME and MSE should be as close to zero as possible [8,40,42,43]. RMSE is useful for comparing interpolation results from different methods; it should be as small as possible. If the ASE is greater than the RMSE, then this indicates that the variability in the dataset has been overestimated; on the contrary, it means that the variability has been underestimated. As for the RMSSE, it should reach a value close to 1 [8,42,43].

Quality analysis of the created interpolation methods will consist of comparing the statistics obtained with their benchmark values, as described above. The approach of creating a smaller test dataset and calculating the estimated values from the interpolation surfaces was also used in comparing the methods. These values were then compared with the measured values at these points. Then, from the estimated points, the mean error or standard deviation can be calculated. Using the RMSE, the sums of similar estimated and measured depth values, calculated mean errors and standard deviations from the test set, a ranking of all methods was created, determining which methods proved to be the best interpolators.

To compare the methods comprehensively, we propose a ranking in which we analyze jointly the above criteria to get a complex response from the research. The ranking is a kind of “averaging” of all the previously mentioned factors, since each of them presents slightly different dependencies, and going by them individually does not make it possible to say unequivocally which method is better than the others. In addition, the ranking will allow verification of the pattern of these statistics—that is, whether one method will be the best or the worst in all statistics, whether half will be the best and the other half the worst, and how this will relate to the final ranking determination. It may turn out here that the best method will be the one for which the statistics were arranged in the middle of the rate of all methods. In the proposed ranking, each method is given a note based on its results calculated with a particular factor. The best method gets 1 and the worst method gets 8, and all the others get numbers sequentially. Finally, the average value for all factors is calculated. The ranking is presented at the end of the analysis in Section 3.2.5.

## 3. Results and Discussion

### 3.1. Initial Data Processing and Analysis

The initial data processing and analysis are essential before proceeding with the interpolation. In this stage, all data information (spatial and non-spatial) was verified and thoroughly analyzed, which is needed to create the best possible models. Without such checking, interpolation results may be unsatisfactory, and it can be difficult to verify the causes of that.

This is also the part in which study limitations related to data take place. These limitations are caused by data acquisition and processing characteristics as well as software requirements, which forced a few assumptions that should be taken into account during data and result analysis. First of all, for the practical reasons described in Section 3.1.1, we assumed that a significant reduction of data is needed prior to further processing. We also decided to remove the outliers, as they significantly affect geostatistical calculations. Additionally, it has to be noted that the study was performed on inland water data, which are considered to be shallow waters in general—this affects the proper analysis of results as the accuracy expected is much higher than in the case of deep waters.

#### 3.1.1. Data Reduction

Before the ESDA could be done, the data had to be reduced. A million-point dataset is not representative (many points overlap), which is important in geostatistical analysis. It is recommended to perform geostatistical analysis on a maximum of 500 points—in the case of larger datasets, the results can be unreliable. The reduction was done on data from orthogonal profiles. The tool used for this was AcrGIS Reduce Point Density, in which the radius of thinning was set at 6.5 m. As a result, the number of points was cut from 5,182,027 to 519. The next step was removing the outliers that could affect the interpolation process. The final reduced layer consists of 494 points (Figure 7). It can be noticed, that Figure 7 shows a gap in the raw data. The gaps can occur in cases of temporary failure of the measurement system (e.g., weak connection with an unmanned vessel). Therefore, in a practical approach, the surveys are conducted at least twice to control the results and ensure full coverage. It was assumed that this gap would not affect the resulting interpolation strongly because there is a large neighborhood of depth estimation points around it.

#### 3.1.2. Exploratory Spatial Data Analysis

The reduced data could be exploratorily analyzed. The goal of this part is to get knowledge about the nature of the dataset, which allows for adjusting better the parameters of geostatistical analysis. It consists in fact of the following steps:6.Visual analysis of plotted data;7.Histogram analysis;8.Normal Q-Q plot analysis;9.Trend analysis.

Firstly, the symbolization of points was changed, dividing them into five depth classes. This allows a general overview of the structure of the data to learn how the depth of the analyzed area is distributed. Figure 8 shows the distribution of depth values in meters.

This step showed that points next to each other are more like each other than those further away, as points in the same depth range formed clusters. Then a histogram analysis was performed (Figure 9).

The created histogram shows a leftward asymmetry of distribution (so the skewness coefficient probably has a negative value). This is evidenced by the fact that, in front of the cluster of bars on the left, there are a few data points whose values deviate significantly from the others. The outliers’ values and locations were checked. As assumed, these are the outlier points with the deepest values (exceeding 25 m). Checking locations is quite important, as their distribution can significantly affect the resulting interpolation surfaces. Figure 10 shows that outliers are concentrated in one place.

Then, we compared this Figure with the points outside the profiles. As a control, the depth values of the points for parallel profiles were checked. It turned out that the depth at this location for parallel profiles was in the range of −6–−7 m, which is much less than 25 m. Thus, it can be considered that these are errors, so they were removed. After removing the histogram, it was checked again. Removal of the erroneous points resulted in the histogram now being positively skewed (Figure 11). The statistics were also checked.

The next tool used in this part was the Normal Q-Q Plot to test whether the data has a normal distribution (Figure 12).

The graph shows that, except for outliers (the deepest and shallowest points), the data tends toward a normal distribution, as most of the data is very close to the baseline (line on 45° angle). This means that spatial data does not have a normal distribution but tends towards it. The normality of the distribution can be disturbed by, among other things, the presence of a trend, which is checked in the next step (Figure 13). Trend analysis allows us to determine the presence or absence of a trend in the input dataset and to determine which order of the polynomial best matches the trend.

From the trend graph, it was clear that a trend is occurring—the green and blue lines are rounding into the shape of a parabola, which indicates a trend of the second degree. Therefore, for further analysis, it was decided that such trends must be included in further modeling.

#### 3.1.3. Structural Analysis and Variogram Modeling

For structural analysis, a semivariogram cloud was created. According to the theory (Tobler’s 1st law of geography) [44,45], the farther the points are on the *x*-axis (that is, pairs of points are farther apart), the farther these points should be from 0 on the *y*-axis. Analysis of the semivariogram cloud showed that as the distance increases, there are fewer and fewer pairs of points that have strong similarities between them (Figure 14).

The trend in the data also determines the occurrence of anisotropy, as the data may not only depend on the distance but also on the direction. For comparison, the clouds were checked for angles direction 0°, 45°, 90° and 135°, i.e., for every possible direction (Figure 15a,b).

Figure 15 show that in the directions 0°, 45° and 90°, the greater the distance, the fewer similar pairs of points there are (although at 0° the number of pairs for most of the graph is similar). The situation is different with a directional angle of 135°—here, with increasing distance, the number of similar pairs also increases. The cloud of the semivariogram at 135° confirms the presence of a trend in the data—the semivariance function increases with further distances. The different level of variation in different directions indicates the presence of spherical anisotropy, so both trend and anisotropy were considered in the modeling variation and estimation stage.

Figure 16 shows the variogram model used for estimation. The determinants of this model were previously studied: trend and anisotropy, as well as the lag size. The selection of a lag size has important effects on the empirical semivariogram. For example, if the lag size is too large, short-range autocorrelation may be masked. If the lag size is too small, there may be many empty bins, and sample sizes within bins will be too small to get representative averages for bins [33]. For the analyzed data, the most optimal lag size is approximately 4.93 m. The experimental variogram followed an exponential model, as shown in Figure 16. The function reaches about 95% of the sill value at a distance equal to the range.

#### 3.1.4. Estimation of Interpolation Surfaces

Implementation of interpolation begins with indicating the target method and dataset (and the attribute of data against which the estimation will be performed). Then follows an optimization of general settings (in the case of deterministic methods) such as type of neighborhood, checking the weights of the points or power of the fit. In the case of geostatistical methods, the next step is modeling the variogram (which is described in Section 3.1.3). Next, cross-validation is performed—all error values and target predicted values are estimated. The final step is creating interpolation maps, which can be visualized and analyzed. Figure 17 presents the created interpolation surfaces for eight methods used. The depth values are in meters. In general, the surfaces presented in the figure are more or less similar. However, deeper visual analysis can lead to important details. The surfaces differ mainly in aspects such as the smoothness of the surface, noticeable in the areas where depth significantly changes; the flatness of the area, noticeable as the size of the medium depth area in the middle and extreme areas in the corners; and the size of local differences, e.g., flattening in the deep area. Taking geostatistical methods into account, it can be seen that UK and EBK give the most varied surfaces. In the UK, the sizes of similar depth areas are rough and the shape of flat areas in the deep part is significantly different than in other methods. This might be the result of the assumption that the unknown local average changes gradually over the study area, which affects global smoothness. EBK, on the other hand, gives the most varied surface in terms of depth changes. Extreme values are boosted, while the medium value area is very small. The gradient is the highest in this method. Thus, the flattening area almost disappeared, and on the other hand, the size of the deepest values area and the size of the smallest depths area is the biggest on all surfaces. This is a result of the local adjustment of the semivariogram to the data within the iterated methodology. Other Kriging examples give relatively similar results; however, DK calculates the most smoothed surface (with the smallest flat area in the deep part) and OK, the least smoothed, but not as rough as the UK method.

Comparison to other methods shows that the geostatistical approach, in general, provides a nice compromise between surface smoothness and nice visual presentation and model accuracy. For example, in the GPI method, the surface is very smooth, but in many places, important changes are lost, like flattening in the deep part. The shape of the depth areas is significantly different than in other methods. On the other hand, the IDW method interpolates values only locally, which results in a good reflection of depth changes; however, the surface is very rough. RBF gives the most similar results to geostatistical methods but is slightly more sensitive to data changes.

### 3.2. Analysis of Results

Interpolation was the final step of geostatistical modeling. Additionally, a deterministic model was created (Figure 14) for comparison. The models were compared according to the verification methodology described in Section 2.4. The plan for the analysis is outlined as follows:Interpolation statistics analysis;Creating a test point dataset;Comparison of measured and estimated depth values (based on the test dataset);Calculation of differences between measured and estimated depth values in the test dataset (mean error, standard deviation);Ranking methods.

The rules for interpretation and the explanation of their meaning for the accuracy analysis are given in Section 2.4; case analysis of values in this section follows the same rules.

#### 3.2.1. Estimation of Interpolation Surfaces

The metrics used for statistical analysis are presented in Section 2.4. In this chapter, we will show these values for interpolation performed by deterministic methods and by Kriging in various variants. Table 1 shows ME and RMSE, which are available for all the methods. Table 2 presents only the interpolation statistics for the geostatistic methods. All values in Table 1 and Table 2 are in meters.

Based on the results, the model values were compared with the following observations:10.In each case, ME is close to 0;11.The MSE value is also close to 0;12.Only in the EBK method, the value of ASE is close to RMSE; in the remaining Kriging methods, the ASE is smaller than the RMSE, which may indicate an underestimation of data variability,13.RMSSE in the EBF method is almost 1, which is almost ideal; in the other cases it exceeds the value of 2, which may indicate an overestimation of data variability,14.Based on RMSE, the ranking of methods is EBK, RBF, DK, OK, SK, IDW, UK, GPI, where GPI clearly stands apart from the others (values 1.7 where other methods have values 1.2–1.3).

#### 3.2.2. Creating a Test Point Dataset

A new class of objects was created with 40 points for interpolation efficiency comparison. An effort was made to indicate points located at different depths and those at the boundaries between two depth classes (Figure 18). This goal might not be achieved using the uniform spatial sampling method. In addition, the method used meets important objectives: equal spatial coverage and equal coverage in feature space.

For the next two analyses, the estimated values for these 40 points are needed. The *GA Layer to Points* tool was run to calculate depth values for these points for each created interpolation surface. The result was eight new layers with estimated depth values for the test area. From these layers, values were extracted, and then, target comparisons and calculations were made.

#### 3.2.3. Comparison of Measured and Estimated Depth Values (Based on the Test Dataset)

Table 3 shows the values of measured and estimated depth for each used method. All values are in meters. Closest to the estimated values are in bold in the Table. The last row of Table shows the number of the best values for each method.

This analysis showed that the most similar estimated values measured is in the IDW method. The next places are followed by EBK, DK, RBF, UK, SK, GPI and OK in last place. The second thing noted is the fact that in this analysis we cannot say if deterministic or geostatistical methods are better interpolators, because there is no significant advantage of one method over the other.

#### 3.2.4. Calculation of Differences between Measured and Estimated Depth Values in the Test Dataset

In the next analysis, the differences between measured and estimated values were calculated. Then, the mean error and standard deviation were calculated. The results are presented in Table 4. Bolded values are the smallest differences between measured and estimated depth values. All values are in meters.

Table 4 shows that the GPI method could create the best interpolation since the mean absolute error is 0.20 m. However, GPI has the highest standard deviation, so the measurement results here are the furthest from the mean, and that fact shatters the suggestion that GPI can be the best interpolator method. A large deviation was also observed for the IDW; additionally, it has the highest average measurement error, which is 0.30 m. The lowest standard deviation, on the other hand, was calculated for the UK method. Generally, in this case, we may conclude that geostatistical methods gave better results because all the Kriging methods have the lowest standard deviation and besides GPI, the lowest mean absolute errors than deterministic methods.

#### 3.2.5. Methods Ranking

The final part of the analysis was preparing a ranking of methods, based on own proposed methodology. The ranking was determined based on the following factors:15.RMSE;16.The sum of most similar observed values between measured and estimated;17.Mean absolute error;18.Standard deviation.

A summary of the parameters that determine the realization of the best interpolation surface is presented in Table 5. Each method was ranked from 1 to 8, where 1 is the smallest RMSE, mean absolute error, standard deviation, and the biggest number of most similar observed values between measured and estimated. Next, all the ranking notes were averaged, and that average decided the final ranking.

The ranking is:Disjunctive Kriging (DK)Empirical Bayesian Kriging (EBK)Simple Kriging (SK)Universal Kriging (UK)Ordinary Kriging (OK) and Radial Basis Function (multiquadric) (RBF)Inverse distance weighted (IDW)Global polynomial interpolation (GPI)

As we can see, the summary of the analysis clearly shows that geostatistical methods are better interpolators for spatial (bathymetric) data.

### 3.3. Verification of the Results with Other Data

The analysis was performed once again in another dataset for verification of results. The dataset was collected in the port of Szczecin, the capital city in Zachodniopomorskie Voivodeship (Figure 19). Like previous steps, the data were reduced (from 634,913 to 467), and the ESDA and structural analysis were performed, then making estimations and analysis of the results.

Figure 20 presents interpolation maps for this area. The depth values are in meters. What is noteworthy is the fact that the EBK method is the only one that does not interpolate the entire area but only a range of data. Visually, we can also see that IDW and OK interpolated different depth values in the middle of the area than the rest of the methods, which basically confirms earlier findings in Figure 17. A similarity of interpolation can be seen on the right side—the region marked in red indicates deeper values. The visual analysis also indicates that the interpolation of the land is due to the neighboring survey points to the land. Of course, the interpolation of land is wrong, because ArcGIS is not able to define on its own what specific area it interpolates, while the researcher himself is aware of this and analyzes only the area of interest (in this case, the water part).

Table 6 shows the results of the analysis including searching most similar points and calculations of mean absolute error and standard deviation for each method.

The analysis of Table 6 shows that, in general, this surface was a harder case to be modeled, as in all cases mean absolute error achieved is higher. This is the result of the land area between the measurement point, which affects interpolation in most cases (all except EBK). Thus, EBK and RBF give good results as the methods adjusted to sudden changes (like between water and land area). Surprisingly good results were obtained with Simple Kriging, which has a significantly bigger number of similar values than the other. The area itself is rather flat, without major gradients of depths (apart from berths), therefore simple model fits well with the data.

The final ranking is presented in Table 7.

Analogies and differences that were observed between the main analysis and the verification analysis are listed below:19.In both cases, the worst methods are IDW and GPI (last place in both final rankings);20.RBF (Multiquadric) in the first analysis was one of the worst methods, in the second one is in the second place;21.In both analyses, the best Kriging methods are Simple, Disjunctive and Empirical Bayesian—in the first case, these methods reached in the ranking respectively third, first and second place, in the second one SK was the best and DK and EBK reached the third place;22.In both cases, the deterministic methods have high standard deviations (especially IDW and GPI).

In general verification research showed that undertaken methodology for interpolation methods assessment is appropriate. RMSE, MAE and standard deviation show qualitative assessment. They are in a large number of cases related to each other. To make the assessment more complete, we have proposed also the sum of the most similar factors, which reflects the quantitative assessment by showing similarity to selected measured data points. Averaging these factors allows us to provide one simple indicator that includes these elements; however, keeping the accuracy (reflected by errors) is the most important.

## 4. Conclusions

The work presents interpolation methods that were subjected to comparative analysis using bathymetric data. In addition to geostatistical methods, which are the main target of the work’s consideration, three deterministic methods were also included in the comparison, assuming that the results of these methods would be inferior to the geostatistical ones. The comparison and assessment of interpolation methods were performed with parameters such as the MAE, standard deviation, and RMSE. However, additionally, a ranking of methods based on the integration of the above parameters was provided. A ranking itself was proposed as an average of statistical factors reflecting an error, similarity, and dispersion. It is, therefore, an attempt to provide a comprehensive assessment in a relatively simple approach. In the future, modifications can be proposed to weigh the factors in other ways or to introduce other indicators.

Based on the results, it can be concluded that more accurate prediction maps were obtained using geostatistical methods (various types of Kriging methods) than deterministic methods. However, the radial basis function does not at all strongly deviate in results from Kriging methods, unlike inverse distance weighted and global polynomial interpolation. Moreover, sometimes it can be better than geostatistical methods. This confirms what was previously established, that RBF is suitable for working with large data sets. The best results were achieved with DK and EBK. Interestingly, those methods are not classical types of Kriging. EBK does not need to consider either anisotropy or trend, which were observed during ESDA spatial and structural analysis, so this raises the question of what is more important for Kriging interpolation—the determination of trend and anisotropy or creating a semivariogram model based on multiple calculations of this function. DK also differs from the classical approaches of the Kriging method, as it is a non-linear type of this method. Nevertheless, for each method, a rather large average error of measurement (range of 0.20–0.30 m for the main analysis and range of 0.30–0.65 m for the verification analysis) was observed. For cartographic purposes, such an error will not be too problematic, but for precise measurements, such values may prove to be too large. However, it should be noted here that the largest errors arise at the extreme locations, where there are a few neighboring points.

The second problem may be related to the reliability of the data—during the ESDA method, several points were captured where the measured values were erroneous. Another issue that should be investigated in the future is the effect of depth on the results—the study described in this paper took place in a fairly shallow and uniform body of water. The next step would be to test a more varied body of water and contrast the results of the uniform bottom analysis with the diverse one. Performing analyses on a wide variety of data will allow a better understanding of the studied data, which is the basis for conducting geostatistical analyses. Further research should also include the analysis of the computational efficiency of interpolation algorithms, taking into account possible implementation in practical applications for bathymetric data modeling.

## Figures and Tables

**Figure 1 sensors-23-03941-f001:**
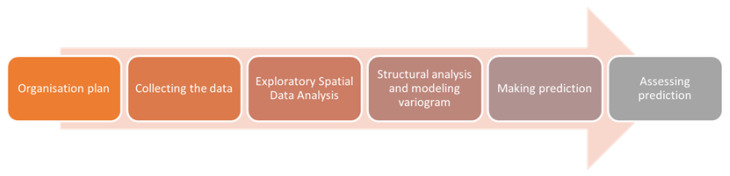
Methodology of geostatistical modeling.

**Figure 2 sensors-23-03941-f002:**
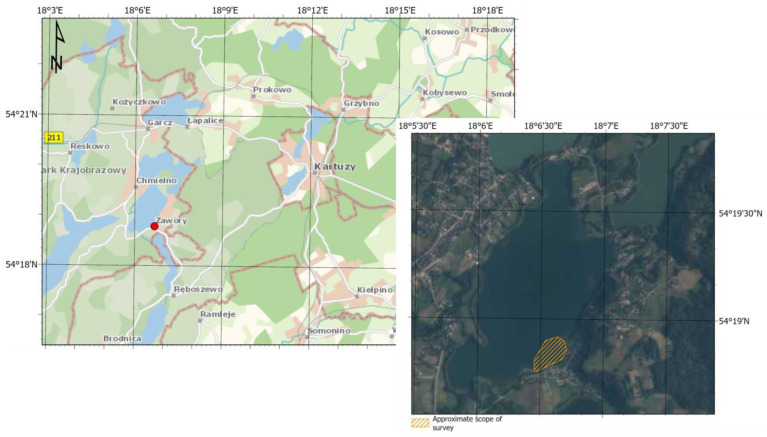
Localization of village Zawory in the topographic map in 1:75,000 scale (**left** side) and area of the bathymetric survey in Lake Klodno in 1:15,000 scale (**right** side).

**Figure 3 sensors-23-03941-f003:**
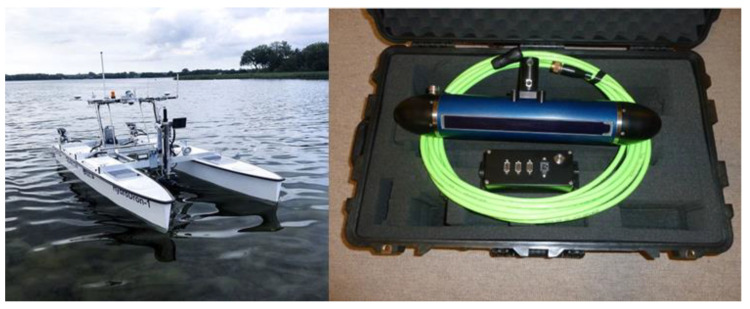
Unmanned vessel HydroDron-1 (**left** side) and interferometric sonar PING3DSS-DX-450 (**right** side).

**Figure 4 sensors-23-03941-f004:**
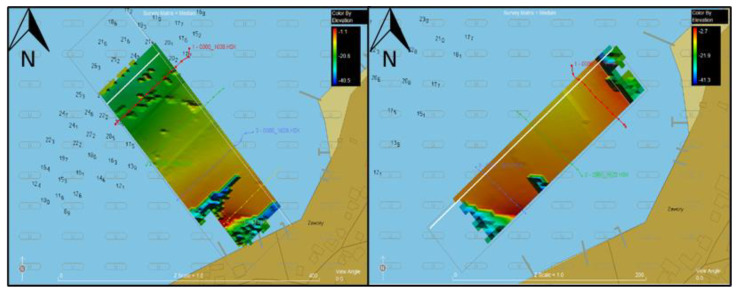
Raw bathymetric data on parallel profiles (**left** side) and perpendicular profiles (**right** side) in HYPACK.

**Figure 5 sensors-23-03941-f005:**
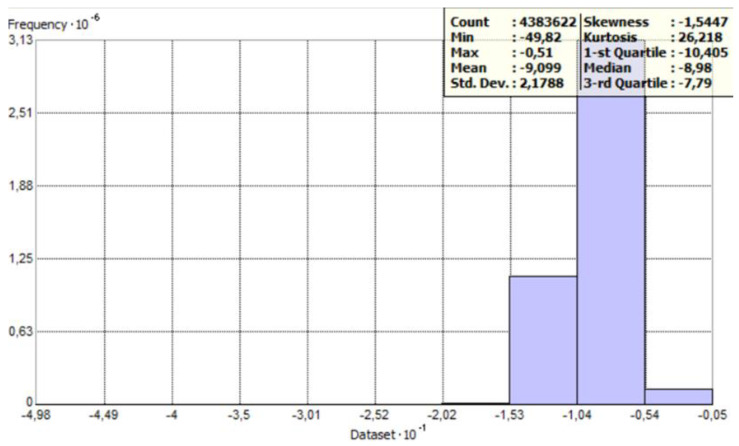
Histogram with statistics for the perpendicular profile—as prepared in ArcGIS software.

**Figure 6 sensors-23-03941-f006:**
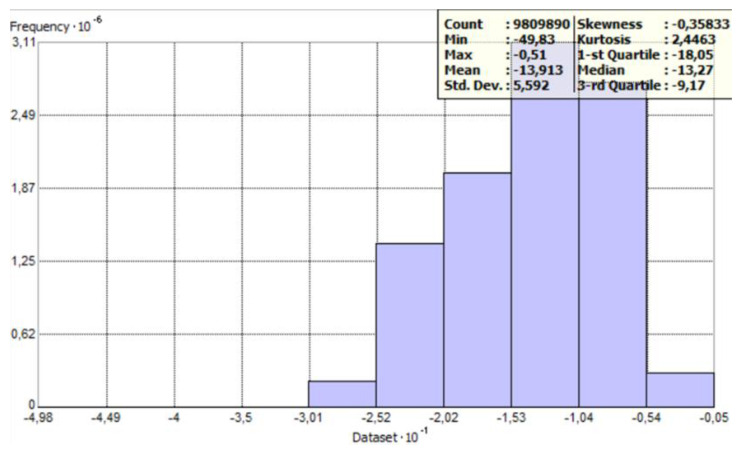
Histogram with statistics for the parallel profile—as prepared in ArcGIS software.

**Figure 7 sensors-23-03941-f007:**
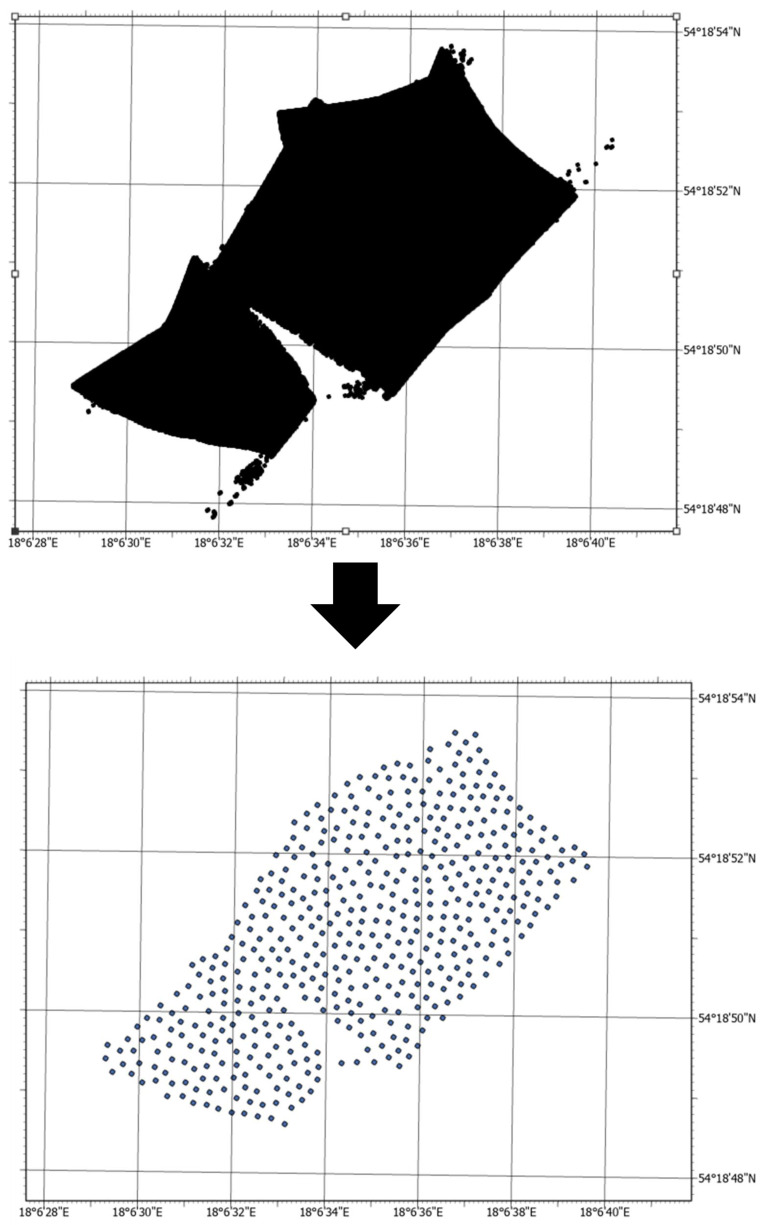
The process of data reduction.

**Figure 8 sensors-23-03941-f008:**
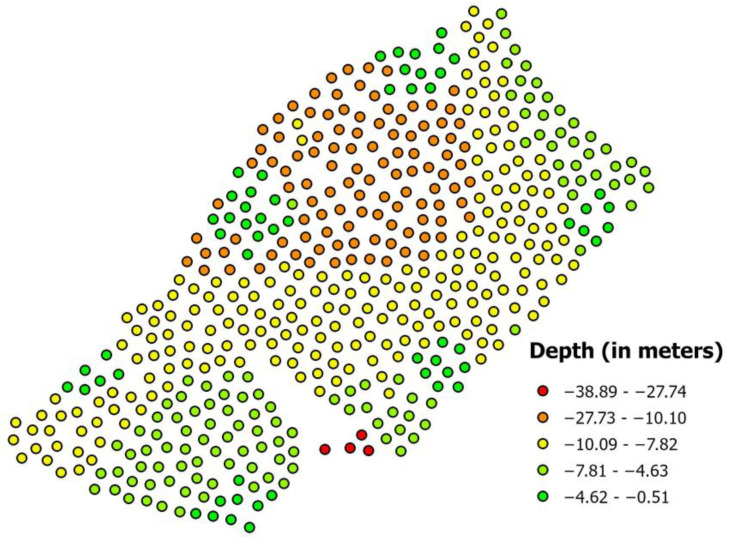
Distribution of five depth classes.

**Figure 9 sensors-23-03941-f009:**
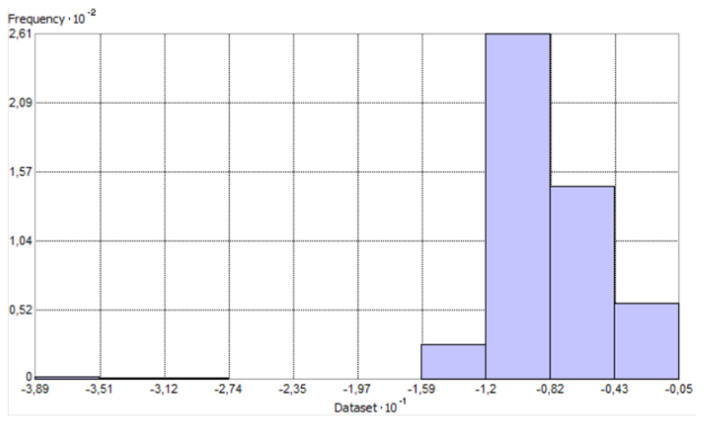
Histogram for analyzed data—as prepared in ArcGIS software.

**Figure 10 sensors-23-03941-f010:**
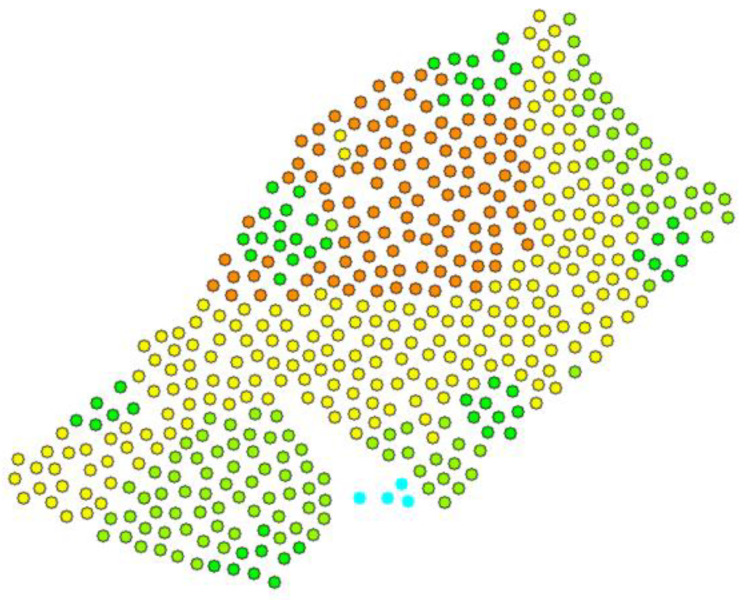
Selected outliers from analyzed data.

**Figure 11 sensors-23-03941-f011:**
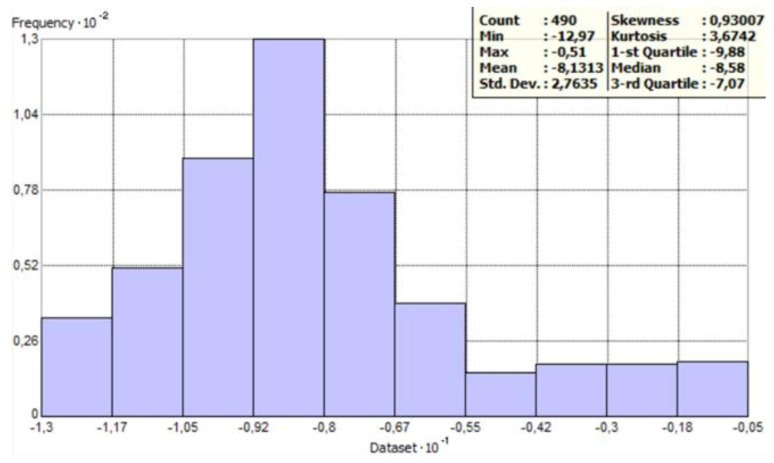
Histogram after removing erroneous points and statistics of the data—as prepared in ArcGIS software.

**Figure 12 sensors-23-03941-f012:**
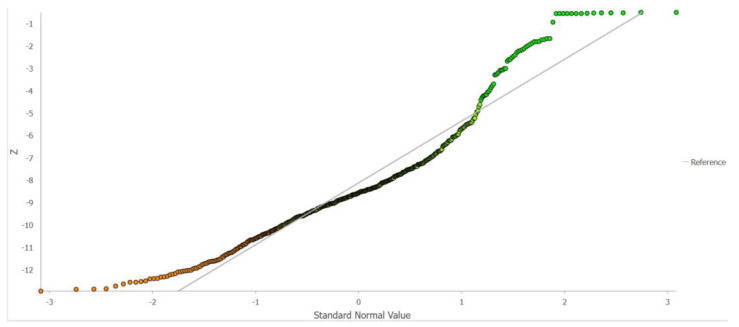
Normal Q-Q Plot.

**Figure 13 sensors-23-03941-f013:**
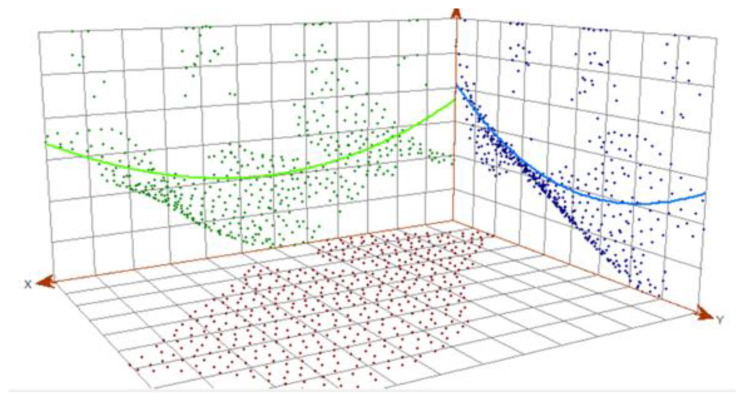
Trend analysis window.

**Figure 14 sensors-23-03941-f014:**
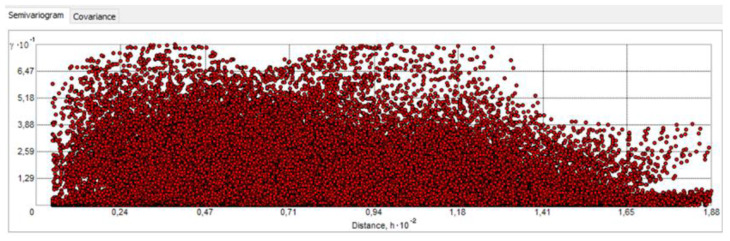
Semivariogram cloud window—as prepared in ArcGIS software.

**Figure 15 sensors-23-03941-f015:**
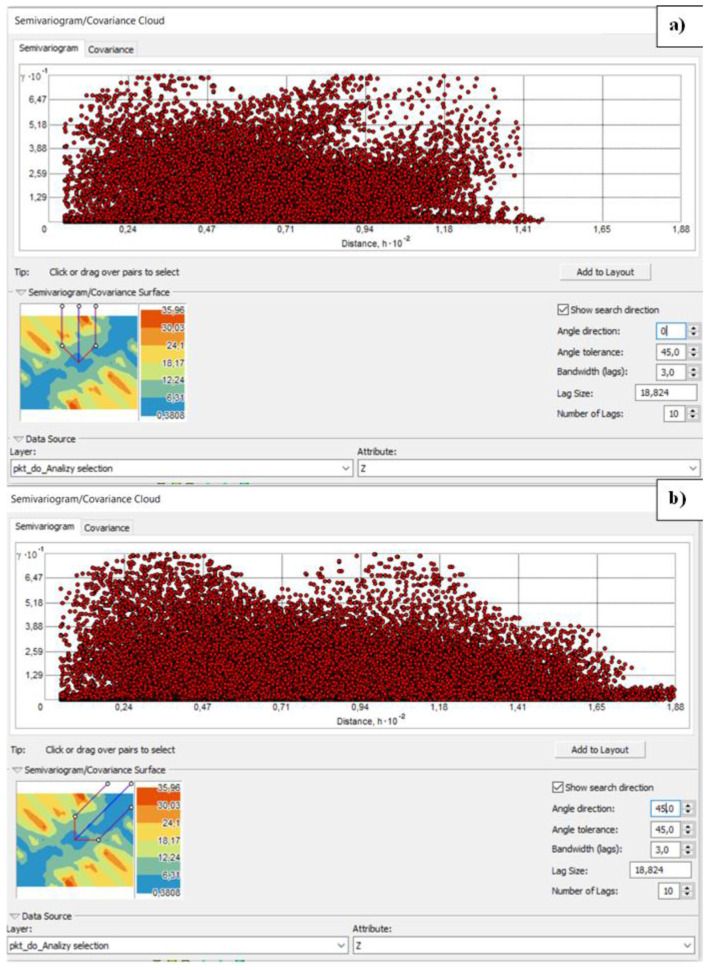
Checking anisotropy in four directions: (**a**) 0°, (**b**) 45°, (**c**) 90°, (**d**) 135°—as prepared in ArcGIS software.

**Figure 16 sensors-23-03941-f016:**
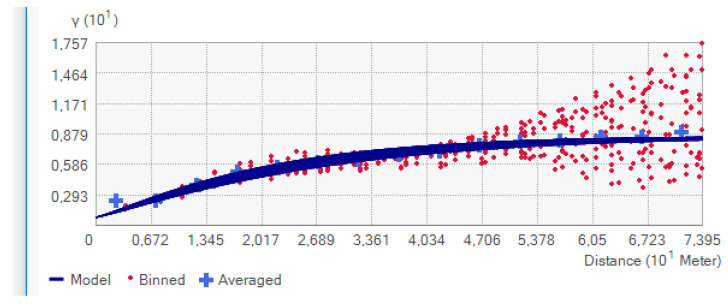
Experimental variogram described by exponential model—as prepared in ArcGIS software.

**Figure 17 sensors-23-03941-f017:**
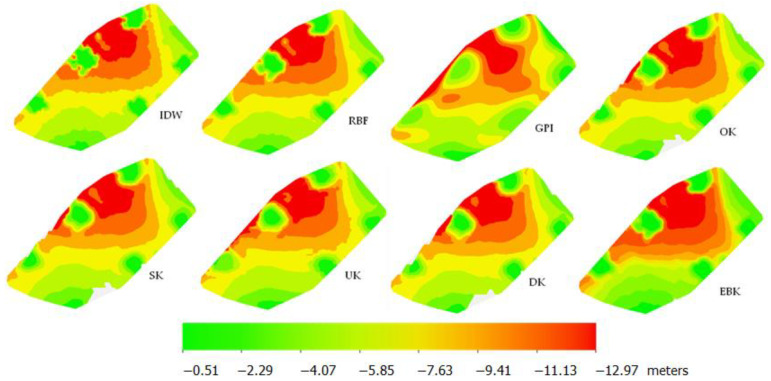
Interpolation maps for all used methods (in meters).

**Figure 18 sensors-23-03941-f018:**
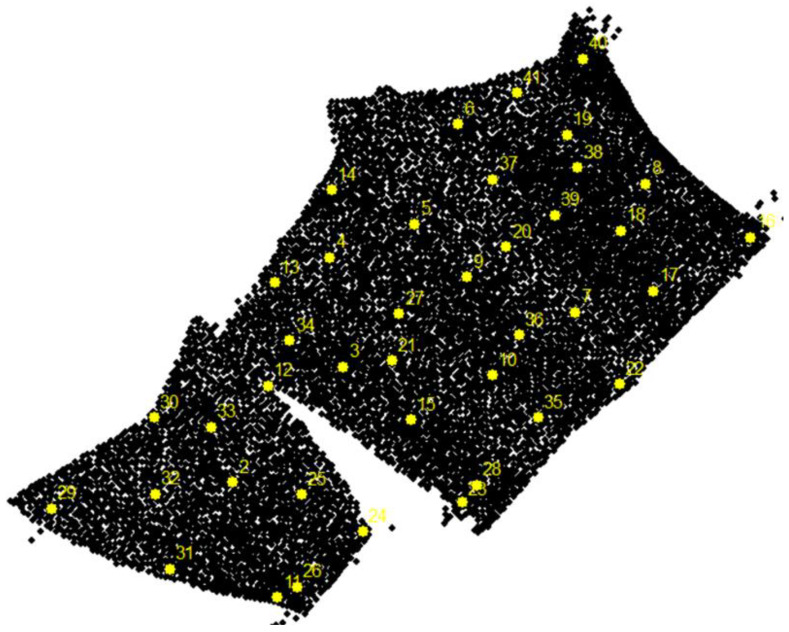
Locations of 40 points of the test dataset.

**Figure 19 sensors-23-03941-f019:**
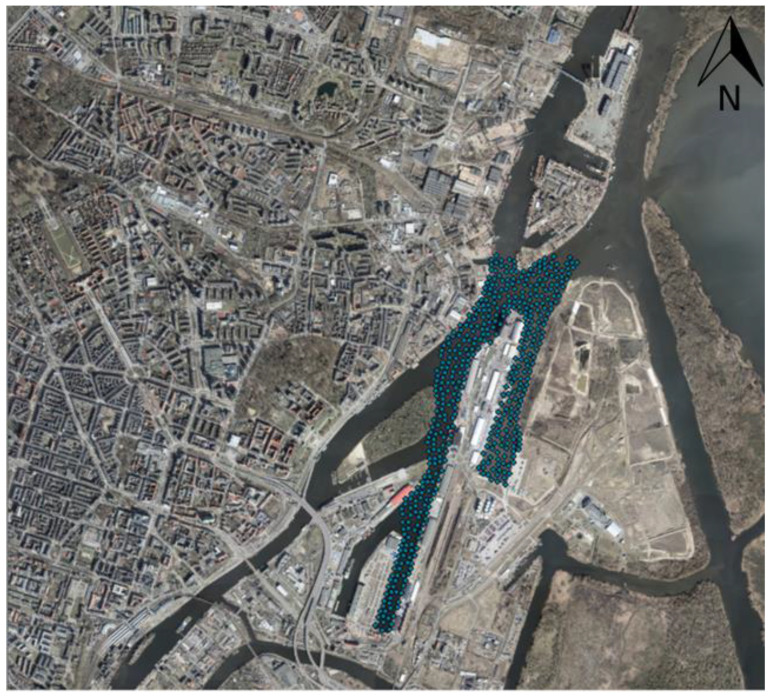
The dataset of port in Szczecin on orthophoto map.

**Figure 20 sensors-23-03941-f020:**
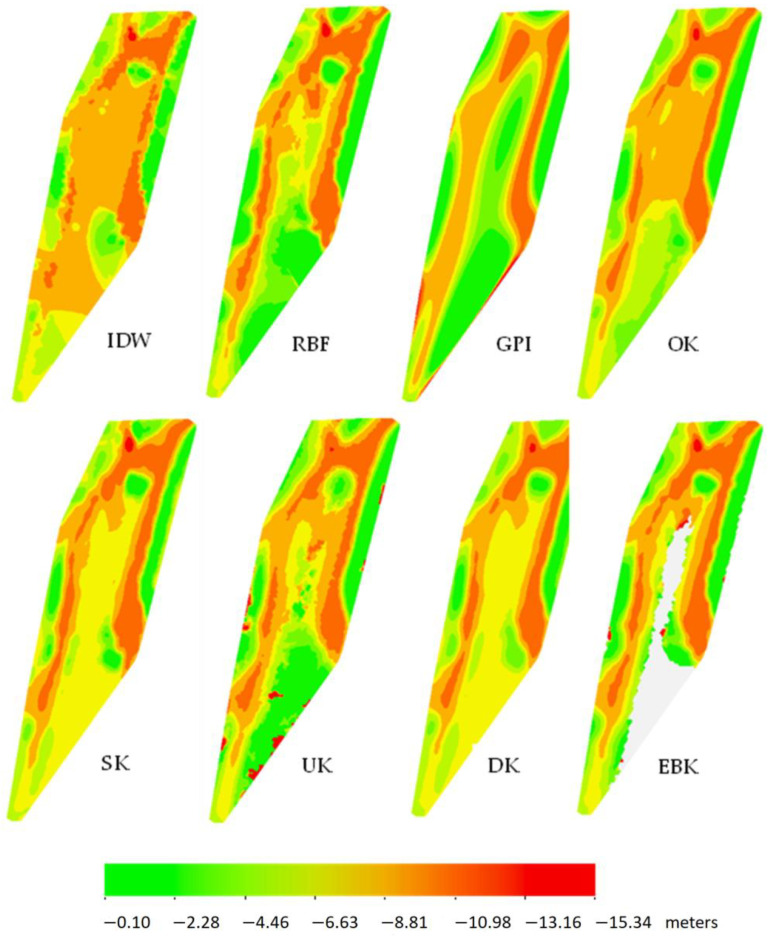
Interpolation maps for the port of Szczecin (in meters).

**Table 1 sensors-23-03941-t001:** Values of ME and RMSE for all used interpolation methods.

	Deterministic Methods			Geostatistic Methods				
	IDW	RBF	GPI	OK	SK	UK	DK	EBK
**ME**	−0.0224	−0.0003	−0.0089	−0.0164	−0.0231	0.0300	−0.0266	−0.0001
**RMSE**	1.3349	1.2723	1.7690	1.3010	1.3016	1.3391	1.2993	1.2584

**Table 2 sensors-23-03941-t002:** Values of MSE, RMSSE and ASE for Kriging methods.

	OK	SK	UK	DK	EBK
**MSE**	−0.0252	−0.0489	0.0389	−0.0309	−0.0181
**RMSSE**	0.6962	1.4848	1.5397	0.7189	0.8945
**ASE**	1.6086	0.8534	0.7408	1.5595	1.1677

**Table 3 sensors-23-03941-t003:** Summary of the measured depth values and estimated depth values in meters.

Point Number	Measured Value	IDW	RBF	GPI	OK	SK	UK	DK	EBK
1	−7.49	−7.32	−7.34	−7.13	−7.35	−7.35	**−7.40**	−7.37	−7.34
2	−9.35	−9.41	−9.36	−9.63	−9.31	−9.33	−9.30	−9.38	**−9.35**
3	−1.06	−0.89	−0.51	−4.66	0.24	−0.19	0.44	−0.07	**−0.93**
4	−11.83	−11.46	−11.38	−11.06	−11.47	−11.51	**−11.55**	−11.52	−11.44
5	−12.32	**−11.95**	−11.68	−9.71	−11.65	−11.15	−10.57	−11.31	−11.55
6	−9.72	**−9.63**	−9.60	−10.19	−9.61	−9.60	−9.62	**−9.63**	−9.61
7	−7.54	**−7.46**	−7.40	−7.97	−7.28	−7.29	−7.28	−7.32	−7.39
8	−10.94	−10.89	−10.89	−11.18	−10.90	−10.89	−10.90	**−10.92**	−10.89
9	−9.11	**−9.14**	−9.06	−9.26	−9.21	−9.22	−9.03	−9.22	**−9.08**
10	−3.08	−3.06	−3.13	−2.89	−3.10	**−3.08**	−2.85	−3.11	−3.13
11	−8.89	−8.61	−8.75	−10.00	−8.79	−8.79	−8.82	**−8.83**	−8.77
12	−10.86	−3.95	−5.11	−5.76	−6.41	−6.45	**−7.74**	−6.36	−5.92
13	−12.30	**−12.34**	−12.38	−8.90	−12.79	−12.25	−11.31	−12.41	−12.12
14	−8.51	−8.55	−8.45	**−8.52**	−8.42	−8.41	−8.41	−8.44	−8.45
15	−5.96	**−5.98**	−5.99	−4.82	−6.31	−6.15	−5.64	−6.25	−5.99
16	−8.42	−8.31	−8.33	−7.34	−7.98	−8.10	−804	−8.13	**−8.35**
17	−8.79	−8.61	−8.74	−8.46	−8.70	−8.72	**−8.80**	−8.76	−8.74
18	−9.87	−9.78	**−9.91**	−8.72	−10.16	−10.02	−10.27	−10.08	−9.93
19	−11.30	−11.11	−11.19	−11.65	−11.19	−11.17	−11.16	**−11.21**	−11.18
20	−9.53	**−9.48**	−9.45	−9.39	−9.42	−9.43	−9.43	−9.46	−9.44
21	−8.56	−8.39	−8.14	**−8.62**	−8.12	−8.12	−8.18	−8.14	−8.13
22	−7.23	−7.26	−7.26	−7.00	−7.22	**−7.23**	−7.29	**−7.23**	−7.26
23	−6.11	−6.07	−6.05	−7.40	−6.09	−6.08	**−6.11**	**−6.11**	−6.08
24	−7.00	−6.96	**−6.99**	−6.85	−6.98	−6.97	−6.93	**−7.01**	−6.98
25	−3.82	**−3.26**	−2.64	−2.56	−2.71	−2.80	−3.17	−2.83	−2.61
26	−10.28	**−10.27**	**−10.27**	−9.81	**−10.29**	−10.32	−10.23	−10.32	**−10.27**
27	−7.58	−7.53	**−7.60**	−5.78	−7.80	−7.76	−7.25	−7.79	−7.68
28	−9.14	**−9.07**	−9.02	−9.60	−8.99	−8.99	−9.03	−9.01	−9.01
29	−2.72	**−2.76**	−3.43	−5.07	−3.70	−3.62	−3.75	−3.53	−3.39
30	−6.66	−6.55	**−6.60**	−6.51	−6.58	−6.57	−6.57	**−6.60**	**−6.60**
31	−8.12	−7.92	−7.94	**−8.07**	−8.21	−8.23	−7.99	−8.22	−7.98
32	−8.51	−8.46	**−8.50**	−7.94	−8.61	**−8.52**	−7.97	−8.56	−8.54
33	−9.86	−9.78	−9.92	−9.57	−10.01	−9.97	−10.10	−10.03	**−9.89**
34	−3.47	−2.73	−1.99	−5.23	−3.26	−3.21	**−3.62**	−3.13	−1.97
35	−9.73	−9.75	−9.75	−10.13	−9.75	**−9.74**	−9.75	−9.77	−9.75
36	−11.84	−11.74	−11.82	−12.14	−11.96	−12.02	−11.77	−12.02	**−11.84**
37	−9.66	−9.68	−9.67	−9.56	−9.60	**−9.66**	−9.79	−9.67	−9.68
38	−10.52	−10.55	**−10.52**	−10.27	−10.46	−10.46	−10.45	−10.49	−10.51
39	−8.81	−8.55	−8.90	−9.26	−9.11	−9.06	−9.25	−9.10	**−8.89**
40	−2.01	−1.21	−0.70	**−1.70**	−0.09	−0.07	−1.21	−0.10	−0.73
** *∑ of the most similar* **		**11**	7	4	1	5	6	8	9

**Table 4 sensors-23-03941-t004:** Calculated differences between measured and estimated depth values, mean average errors and standard deviations in meters.

Point Number	Measured Value	IDW	RBF	GPI	OK	SK	UK	DK	EBK
1	−7.49	0.17	0.15	0.36	0.14	0.14	**0.09**	0.12	0.15
2	−9.35	−0.06	−0.01	−0.28	0.04	0.02	0.05	−0.03	**0.00**
3	−1.06	0.17	0.55	−3.60	1.30	0.87	1.50	0.99	**0.13**
4	−11.83	0.37	0.45	0.77	0.36	0.32	**0.28**	0.31	0.39
5	−12.32	**0.37**	0.64	2.61	0.67	1.17	1.75	1.01	0.77
6	−9.72	**0.09**	0.12	−0.47	0.11	0.12	0.10	0.09	0.11
7	−7.54	**0.08**	0.14	−0.43	0.26	0.25	0.26	0.22	0.15
8	−10.94	0.05	0.05	−0.24	0.04	0.05	0.04	**0.02**	0.05
9	−9.11	−0.03	0.05	−0.15	−0.10	−0.11	0.08	−0.11	**0.03**
10	−3.08	0.02	−0.05	0.19	−0.02	**0.00**	0.23	−0.03	−0.05
11	−8.89	0.28	0.14	−1.11	0.10	0.10	0.07	**0.06**	0.12
12	−10.86	6.91	5.75	5.10	4.45	4.41	**3.12**	4.50	4.94
13	−12.30	**−0.04**	−0.08	3.40	−0.49	0.05	0.99	−0.11	0.18
14	−8.51	−0.04	0.06	**−0.01**	0.09	0.10	0.10	0.07	006
15	−5.96	**−0.02**	−0.03	1.14	−0.35	−0.19	0.32	−0.29	−0.03
16	−8.42	0.11	0.09	1.08	0.44	0.32	0.38	0.29	**0.07**
17	−8.79	0.18	0.05	0.33	0.09	0.07	**−0.01**	0.03	0.05
18	−9.87	0.09	**−0.04**	1.15	−0.29	−0.15	−0.40	−0.21	−0.06
19	−11.30	0.19	0.11	−0.35	0.11	0.13	0.14	**0.09**	0.12
20	−9.53	**0.05**	0.08	0.14	0.11	0.10	0.10	0.07	0.09
21	−8.56	0.17	0.42	**−0.06**	0.44	0.44	0.38	0.42	0.43
22	−7.23	−0.03	−0.03	0.23	0.01	**0.00**	−0.06	**0.00**	−0.03
23	−6.11	0.04	0.06	−1.29	0.02	0.03	**0.00**	**0.00**	0.03
24	−7.00	0.04	**0.01**	0.15	0.02	0.03	0.07	**−0.01**	0.02
25	−3.82	**0.56**	1.18	1.26	1.11	1.02	0.65	0.99	1.21
26	−10.28	**0.01**	**0.01**	0.47	**−0.01**	−0.04	0.05	−0.04	**0.01**
27	−7.58	0.05	**−0.02**	1.80	−0.22	−0.18	0.33	−0.21	−0.10
28	−9.14	**0.07**	0.12	−0.46	0.15	0.15	0.11	0.13	0.13
29	−2.72	**−0.04**	−0.71	−2.35	−0.98	−0.90	−1.03	−0.81	−0.67
30	−6.66	0.11	**0.06**	0.15	0.08	0.09	0.09	**0.06**	**0.06**
31	−8.12	0.20	0.18	**0.05**	−0.09	−0.11	0.13	−0.10	0.14
32	−8.51	0.05	**0.01**	0.57	−0.10	**−0.01**	0.54	−0.05	−0.03
33	−9.86	0.08	−0.06	0.29	−0.15	−0.11	−0.24	−0.17	**−0.03**
34	−3.47	0.74	1.48	−1.76	0.21	0.26	**−0.15**	0.34	1.50
35	−9.73	−0.02	−0.02	−0.40	−0.02	**−0.01**	−0.02	−0.04	−0.02
36	−11.84	0.10	0.02	−0.30	−0.12	−0.18	0.07	−0.18	**0.00**
37	−9.66	−0.02	−0.01	0.10	0.06	**0.00**	−0.13	−0.01	−0.02
38	−10.52	−0.03	**0.00**	0.25	0.06	0.06	0.07	0.03	0.01
39	−8.81	0.26	−0.09	−0.45	−0.30	−0.25	−0.44	−0.29	**−0.08**
40	−2.01	0.80	1.31	**0.31**	1.92	1.94	0.80	1.91	1.28
** *Mean absolute error* **		0.30	0.30	**0.20**	0.23	0.25	0.26	0.23	0.28
** *Standard deviation* **		1.09	0.97	1.42	0.83	0.81	**0.66**	0.82	0.85

**Table 5 sensors-23-03941-t005:** Ranking of the interpolation methods.

	IDW	RBF	GPI	OK	SK	UK	DK	EBK
**RMSE**	6	2	8	4	5	7	3	**1**
**∑ of the most similar**	**1**	3	6	7	5	4	3	2
**Mean absolute error**	6	6	**1**	2	3	4	2	5
**Standard deviation**	7	6	8	4	2	**1**	3	5
**Average**	5	4.25	5.75	4.25	3.75	4	2.75	3.25

**Table 6 sensors-23-03941-t006:** Results of analysis in verification analysis.

	IDW	RBF	GPI	OK	SK	UK	DK	EBK
**∑ of the most similar**	5	6	2	4	10	6	6	3
**Mean absolute error**	−0.64	−0.31	−0.49	−0.40	−0.33	−0.46	−0.34	−0.39
**Standard deviation**	0.80	0.52	1.09	0.57	0.42	0.75	0.57	0.43

**Table 7 sensors-23-03941-t007:** Ranking of the interpolation methods in verification analysis.

	IDW	RBF	GPI	OK	SK	UK	DK	EBK
**RMSE**	8	2	7	4	**1**	6	5	3
**∑ of the most similar**	3	2	6	4	**1**	2	2	5
**Mean absolute error**	8	**1**	7	5	2	6	3	4
**Standard deviation**	6	3	7	4	**1**	5	4	2
**Average**	6.25	2	6.75	4.25	1.25	4.75	3.5	3.5

## Data Availability

The data presented in this study are available on request from the corresponding author. The data are not publicly available due to internal ownership rules.

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
