# Peer review of "Comparative Analysis of Selected Geostatistical Methods for Bottom Surface Modeling"

_sensors, 2023, doi:10.3390/s23083941_

Round 1

Reviewer 1 Report

Comments and Suggestions for Authors

The study presents a detailed comparative analysis of the ranges of interpolation methods, validates them statistically based on selected metrics, identifies the best-performing approach, and ranks them accordingly. The authors explain the methods used to generate interpolation surfaces, including the software used, methodological briefs on the optimization process and steps, and visualizations of the results. They highlight the suitability of the proposed methods for handling large datasets, which is crucial for practitioners and researchers working with big data. I highly recommend that the paper be published. However, I have some minor and major comments and suggestions that the authors should consider to improve the quality of the manuscript. These will be presented in detail as follows:

1.       Abstract: In the abstract, the authors should include a sentence that provides specific details about the study. This may include information about the data and characteristics, comparing interpolation methods and parameters to assess the accuracy, preferably with some quantitative results. Information about the practical implications of the study's results and their real-world applications should also be added.

2.       Introduction:  The introduction is too long and may cause readers unfamiliar with this topic to lose interest before getting to the main content. Additionally, the authors did not emphasize knowledge gaps or gaps in existing literature or clarify the research questions they set to answer. I hope the authors will rework this section to make it more organic and straightforward than the current version.

3.       In Figure 2, the labels written on the open-source image in the left corner are invisible and must increase their readability. Additionally, adding grids and graticules to illustrate the location of the study area in Poland would be beneficial.

4.       Line 194-196, the sentences In this work we decided to use also selected deterministic methods, apart of the geostatistical methods. Deterministic methods are often compared with geostatistical to indicate whether geostatistical methods perform better for spatial data.” is appeared to be a bit vague. Please add a reference (s) that supports this claim. Also, rewrite the sentence “These statistics will be compared to the model values described above in the analysis of the created interpolation surfaces to determine the quality of these surfaces” in lines 317-318 for a better explanation.

5.       Results and Discussion:

·         The section is well-organized but lacks an in-depth discussion of the results. The authors present tables of statistical values and figures generated by the proposed method, but they do not provide a detailed interpretation of these values or an explanation of their implications regarding the accuracy of the interpolation. Precise clarification of the values and their significance is needed to improve this section. Moreover, explaining why specific interpretation techniques performed better or worse than others would be helpful.

·         In line 346, the “Exploratory Spatial Data Analysis” is abbreviated before; once the full description of the acronyms was defined at first in their appearance in the main text, they can be used consistently. There are other places like this in this manuscript; please check them throughout the paper and correct them accordingly.

·         There were also some places in which the methods appeared/mixed with that of the “Results and Discussion”; e.g., in lines, 432-433, “Estimates of interpolation surfaces were made using ArcGIS software, which contains Geostatisical Analyst extension dedicated for these analyses.” should be either removed or moved and merged in subsection 2.3.

·         The manuscript provides information regarding the spatial distribution of the test points, which is crucial for validating the accuracy of various interpolation techniques. However, the paper lacks information concerning the study's limitations, such as the assumptions taken into account during data analysis and their potential impact on the results. Including this information would help obtain a comprehensive understanding of the results' accuracy.

·         The authors only consider one type of data (bathymetric data), so the generalizability of their findings to other types of data is uncertain. Furthermore, the study does not provide information about the computational efficiency of the different interpolation techniques, which may be necessary to test the effectiveness of the proposed method on various types of data for practical applications.

6.       Conclusions: What limitations (or gaps not addressed by this study) need further investigation, and how can underlying challenges in this research be overcome?

Author Response

Dear Reviewer,

Thank you for your Review.

Please see our answers in the attachment.

Kind regards

Authors

Reviewer 2 Report

 This research made a comparative analysis of selected geostatistical methods for bottom surface modeling. Overall, it is well organized and maybe of interest of relevant scientists, managers and even software developers. However, some revisions are required before acceptance for publication.

1. The writing needs to be improved so that it can be more understandable. For example, Line 27, the term "bottom of a water region" needs to be explained. Line 66, why is the word "The" capitalized? The reference format used on Line 103, such as "In [26]," is not recommended. Words like "subchapter" in the paper should be changed to "section." Line 386, what does "in the in the" mean? Please check the writing thoroughly the full text.

2. Figure 2, it is difficult to see the location of the village in Poland. Please add the longitude and latitude in the map. The placenames in the map are also unclear. You can delete most of the minor placenames and only keep some important placenames. In the right panel, please give a legend for the orange shade.

3. Lines 174-177, please provide the spatial resolution of the observation.

4. In Figure 7, why are there gaps in the sampling positions? Please briefly explain the impacts of these gaps on the experiment. Please also add UTM coordinates.

5. Lines 477-478, why not use a uniform spatial sampling method? What is the difference in experimental statistics between the method used in this paper and a uniform sampling method?

6. Table 3 shows that the best interpolation method differs from the method described in the paper. For example, for Point number 35, the bold number is not the "closest" one. Also, please check whether the commas “,” in the entire paper should be decimal points ".".

7. In Section 3.2.5, the paper uses four statistical methods to evaluate the performance of different interpolation methods. The standard deviation measures the dispersion of estimated values, while the other three methods calculate the errors between observed and estimated values. Does taking the arithmetic mean of these two types of statistics overestimate the "error"? I believe this method cannot accurately express the quality of interpolation, so please provide an explanation.

8. In Section 3.3, the observed data station surrounds a land area. From Figure 19, it can be seen that the interpolation process does not assign values to the land area. However, would the data on the east side of the land area affect the interpolation results on the west side? This is logically flawed and needs to be addressed. Please explain the impact of land on the interpolation.

9. Some references in the reference list lack representativeness and were published on lower-level journals. In general, citing references from high-level journals can help enhance the impact of your paper. It would be acknowledged if the following papers can be cited, which also used the Kriging method and are closely related to the interpolation techniques in this manuscript.

Xie, D.; Wang, Z. B.; Huang, J.; Zeng, J. River, tide and morphology interaction in a macro-tidal estuary with active morphological evolutions. Catena, 2022, 212, 106131.

Author Response

(The authors gave the same response as above.)

Reviewer 3 Report

The manuscript is well written but the authors do not discuss the details of variogram and variogram models which are essential for estimation. The authors have identified the anisotropy direction using a semi-variogram cloud window. However, the authors do not show the experimental semi-variogram and how the semi-variogram model was fitted. The authors do not discuss the model used for estimation (Such as exponential variogram, spherical variogram, or Gaussian variogram models). The authors do not discuss the nugget, sill, and ranges along the major and minor axis that determines the search ellipse and the parameters for the estimation. 

Author Response

(The authors gave the same response as above.)

Round 2

Reviewer 1 Report

Dear Authors,

I appreciate the effort you have put into revising your manuscript. I am pleased to inform you that your work is now suitable for publication. However, a small matter needs your attention: the grids and graticule on the maps in Figure 7 need to be clarified for better readability.

Once again, congratulations on your valuable contribution to the field.

Best regards,

Author Response

Dear Reviewer,

Thank you for your valuable review.

Please see the attachement with answer and the revised manuscript.

Kind regards

Authors

Reviewer 2 Report

The authors have treated almost all my comments suitable. I have not further comments or suggestions now.

Author Response

Dear Reviewer,

Thank you for your valuable review.

Please see the attachement with answer.

Kind regards

Authors

Reviewer 3 Report

The authors have provided a repose to my comments. The additions they have provided variograms that have the pure nugget effect. The results kriging with a pure nugget effect model means that all weights are equal, and the estimation produced becomes a simple average of the available sample data. The kriging is an important part of this manuscript and the author needs to fix the variograms and figure out the spatial continuity to get proper kriging estimates before the manuscript can be accepted. 

Author Response

(The authors gave the same response as above.)

Round 3

Reviewer 3 Report

The authors have put a lot of effort to improve the manuscript and the manuscript can be accepted in its current form. I am happy with the changes they have made and I do not have any further comments.